# Effectiveness of a Psycho-Social Intervention Aimed at Reducing Attrition at Methadone Maintenance Treatment Clinics: A Propensity Score Matching Analysis

**DOI:** 10.3390/ijerph16224337

**Published:** 2019-11-07

**Authors:** Xiaoyan Fan, Xiao Zhang, Huifang Xu, Fan Yang, Joseph T.F. Lau, Chun Hao, Jinghua Li, Yuteng Zhao, Yuantao Hao, Jing Gu

**Affiliations:** 1Department of Medical Statistics, School of Public Health, Sun Yat-sen University, Guangzhou 510080, Guangdong, China; fanxy3@mail2.sysu.edu.cn (X.F.); 20191007@zcmu.edu.cn (X.Z.); haochun@mail.sysu.edu.cn (C.H.); lijinghua3@mail.sysu.edu.cn (J.L.); Haoyt@mail.sysu.edu.cn (Y.H.); 2Zhejiang Chinese Medical University, Hangzhou 310053, Zhejiang, China; 3Guangzhou Centre for Disease Prevention and Control, Guangzhou 510080, Guangdong, China; xuhuifang1027@21cn.com (H.X.); azbfzk@gzcdc.org.cn (Y.Z.); 4Institute for Global Health and Infectious Diseases, University of North Carolina, Project-China, Guangzhou 510080, Guangdong, China; fan.yang@med.unc.edu; 5Centre for Medical Anthropology and Behavioural Health, Sun Yat-sen University, Guangzhou 510080, Guangdong, China; jlau@cuhk.edu.hk; 6Centre for Health Behaviours Research, School of Public Health and Primary Care, Faculty of Medicine, The Chinese University of Hong Kong, Hong Kong, China; 7Sun Yat-sen Global Health Institute, Sun Yat-sen University, Guangzhou 510080, Guangdong, China

**Keywords:** methadone maintenance treatment, psycho-social intervention, attrition, propensity score matching, China

## Abstract

Methadone maintenance treatment (MMT) is an important approach to address opioid dependence. However, MMT clinics usually report high attrition rates. Our previous randomized controlled trial demonstrated additional psycho-social services delivered by social workers could reduce attrition rates compared to MMT alone. This study aimed to evaluate the effectiveness of psycho-social service in a real-world context. A quasi-experimental design and propensity score matching was adopted. 359 clients were recruited from five MMT clinics in Guangzhou from July 2013 to April 2015. One 20-minute counseling session was offered to the control group after enrolment. The intervention group received six sessions of psycho-social services. The baseline characteristics were unbalanced between two arms in the original sample. After propensity score matching, 248 participants remained in the analysis. At month six, the intervention group had a lower attrition rate [intervention (39.5%) versus control (52.4%), *P* = 0.041], higher monthly income [monthly income of 1000 CNY or higher: intervention (55.9%) versus control (39.0%), *P* = 0.028)], higher detoxification intention score [full intention score: intervention (51.6%) versus control (32.5%), *P* = 0.012)], higher family support in MMT participation [intervention (77.9%) versus control (61.4%), *P* = 0.049)]. This study demonstrated that psycho-social services delivered by social workers can reduce MMT clients’ attrition and improve their well-being in real-world settings.

## 1. Introduction

By the end of 2017, there were 40.48 million people with opioid dependence, accounting for 55% of the total number of people with a drug use disorder globally. [1]. Methadone, as a weak-opioid agonist, has been extensively adopted as a substitution treatment for opioid dependence [2]. When administered daily, methadone can alleviate withdrawal symptoms and reduce an addicted individual’s cravings and drug-seeking [3]. Methadone maintenance treatment (MMT) has been proven effective in terms of retaining patients, suppressing heroin use [4] and reducing needle sharing and HIV transmission [5,6]. Daily methadone use can also reduce criminal activity and improve employment rates and social well-being [7,8].

Despite these advantages, suboptimal retention in MMT programs remains a significant concern worldwide. Data from Ireland [9], Israel [10,11], Tanzania [12] and Iran [13] indicated 1-year MMT retention rates ranging from 34.4% to 76.2%. Similar problems have been observed in MMT programs in China [14]. One meta-analysis including 43,263 Chinese MMT participants reported 1-year retention rate of 55.2% [95% confidence interval (CI): 48.5–61.7%] [15]. Studies conducted in different cities reported retention rates varying from 30.0% in Shanghai at month six to 70.3% in Xi’an at month 12 [16].

Ancillary services, especially psycho-social services, are recommended for the improvement of MMT retention and treatment outcomes [17,18,19,20]. A meta-analysis of 12 studies suggested that the addition of a psycho-social intervention to standard-of-care MMT service could reduce heroin use among clients (relative risk(RR) = 0.69, 95% CI: 0.53–0.91) [21]. One study conducted in Xi’an reported the efficacy of additional psychological counseling during MMT in terms of reducing illicit drug use (RR = 0.534, 95% CI: 0.408–0.700) [22]. Furthermore, ancillary psycho-social services can improve adherence and MMT-related knowledge [23], reduce substance use [24] and ameliorate mental health issues [25].

In China, current MMT guidelines strongly recommended the provision of ancillary services to clients. However, discrepancies have been observed in the types and contents of the services delivered by different clinics [26,27]. Furthermore, the medical staff members at MMT clinics are overburdened with clinical tasks and lack relevant mental health training [28], and therefore, are not the best candidates for the provision of ancillary services. Notably, Hong Kong first introduced social workers to provide psychological services to MMT clients during the early 1990s [29]. Since then, progress has been made in terms of promoting clients’ health status, increasing employment and improving family relationship to support MMT use.

The Guangdong province is estimated to have 457,000 registered drug users, or one sixth of all users in China [30]. The first set of MMT clinics was established in the province in January 2006. Three of these were set up in Guangzhou, the capital city of Guangdong. Subsequently, psycho-social services delivered by social workers were introduced in Guangzhou in 2009, following the promising improvements in social services in Hong Kong. In our previous randomized controlled trial (RCT) of three clinics, we demonstrated that the combination of standard MMT with behavioral theory-based, social worker-delivered psycho-social services effectively reduced the probability of attrition by 18% after six months [31]. These encouraging results led the local government to fund these services and expand the program to more clinics.

In daily practice, social workers select and provide services to clients in a non-random manner since they believe clients with greater needs will benefit more from these services. This study aimed to evaluate the effectiveness of the services in real-world settings, following the demonstration of the efficacy in the previous RCT. Quasi-Experimental study design and propensity score matching (PSM) were used to evaluated differences in attrition and psycho-social status after balancing potential confounding factors between the intervention and control groups.

## 2. Materials and Methods

### 2.1. Study Design and Participants

This research adopted a two-arm, non-blinded quasi-experimental design and was conducted at six MMT clinics in Guangzhou between July 2013 and October 2015. Each clinic employed at least one social worker to provide services to 80–300 MMT clients. MMT clients who met the following criteria were deemed eligible to participate: 1) aged 18 or above; 2) active heroin user prior to admission; 3) registered permanent or temporary resident of Guangzhou and 4) willing to provide written informed consent.

According to the practical manual for social workers in the MMT clinics [31], participants who were more socioeconomically vulnerable (e.g., female, youth), faced a higher risk of relapse or were actively seeking help should be prioritized to receive psycho-social services from trained social workers. From July 2013 to April 2015, MMT clients who received psycho-social services delivered by social workers were classified as the intervention group. Other MMT clients received no psycho-social service were regarded as the control population. In this study, after recruiting one participant to the intervention group, the social worker invited the next client attending the clinic, who did not have the prioritized characteristics, to the control group. The 6-month psycho-social services were delivered to MMT clients of the intervention group by social workers after their enrolment. The study was approved by the Ethics Committee of the School of Public Health, Sun Yat-sen University.

### 2.2. Procedure

Social workers evaluated and approached prioritized eligible clients after reviewing the clients’ treatment profile. They then introduced the program and asked about each client’s intention to participate in the study. During recruitment, the social workers assured the clients that their participation was voluntary, data would be kept confidential and the choice to participate or not would not affect their access to services provided at the clinics. Participants in the control group were introduced, approached and assured in the same way. After providing informed consent, all participants completed baseline questionnaires. Follow-up questionnaires were scheduled at month one and month six. Both baseline and the follow-up questionnaires were completed via face-to-face interviews conducted by trained investigators in a private setting. A flow chart of the procedure is shown in Figure 1.

### 2.3. Intervention

#### 2.3.1. Intervention Group

The intervention of this study was designed based on our previous RCT grounded in the Behavioral Maintenance Theory [31]. Five individual sessions and one family session were delivered to the intervention group over the six months after enrolment. The sessions were divided into three phrases, each with its specific objectives. The first phase had a duration of one month and included two individual sessions designed to introduce MMT and clinics, establish rapport and correct misconceptions about MMT [32]. An additional family session to improve familial communication and support for clients was arranged according to the availability of the participants’ family members. During the second phase, which included months two and three, one session focused on increasing self-efficacy of maintained MMT usage. Clients were asked to recall improvements in their physical health and reductions in drug dependence with the intent to strengthen their confidence. The last phase included months four to six and comprised two individual sessions intended to warn clients about the contextual factors conducive to relapse and the consequences of relapse, reinforce their motives and promote a healthy lifestyle. Apart from the family session, other psycho-social service was provided in individual sessions.

#### 2.3.2. Control Group

After providing informed consent, the control group participants received a 20-minute counseling session related to MMT from clinicians. The contents of the session covered basic knowledge of MMT and answers to the participants’ questions (if any). Social workers’ services to the control group were only available upon specific request and no structured counseling (as in the intervention group) was provided.

### 2.4. Measures

The primary outcome was the attrition rate at month six. Attrition was defined as a failure to visit the MMT clinic for at least seven consecutive days from July 2013 to October 30, 2015. Retention was defined as a lack of attrition during the study. The attrition rates were also calculated using 1- and 12-month data. The secondary outcomes were based on questionnaire questions, including employment status, monthly income, current drug use and detoxification intention score, MMT-related perceptions, social support, family relationship to support MMT use and health status at month six.

The questionnaire was based on previous research and literature review and was finalized after a pilot study of 20 participants. Sociodemographic variables (gender, age, education, marital status, HIV serostatus), drug use-related characteristics (age of initial drug use, duration of drug use, drug injection history, times of compulsory detoxification) and MMT-related characteristics (willingness to receive psycho-social service and their self-efficacy to retain in MMT in the next six months) were only measured during the baseline survey. Variables including socioeconomic status (employment status and income), current drug use and detoxification intention score, MMT-related perceptions, family and social support, health status, were measured at both baseline and the follow-up questionnaires scheduled at month one and month six. The methadone-use histories at month 1, 6 and 12 were retrieved from the information systems of MMT clinics.

#### 2.4.1. Drug Use and Detoxification Intention

Participants were asked about the frequency of drug use during the previous month and their current detoxification intention score (measured using a 0–10-point scale, with 0 representing no and 10 representing full intention). The intention scores were dichotomized as either full intention (score = 10) or other.

#### 2.4.2. MMT-Related Perceptions

Ten true-or-false items were used to measure the MMT-related perceptions from two distinct aspects. Four items assessed perceptions about the aim and duration of MMT, while six addressed the MMT dosage [31,32]. The clients were given statements such as ‘MMT requires life-long methadone use’ and ‘methadone is harmful to our health and we shouldn’t use too much of it’ and asked to judge whether they were true. Participants with a ≥50% correct response rate to each aspect of MMT were identified as having a proper perception of the aim/duration of MMT or dosage.

#### 2.4.3. Family and Social Support

Perceived social support was measured using a four-item Social Support Scale [33], which included questions such as ‘How much substantial help could you obtain from your friends if you encountered difficulties in life?’ The response options varied from one (‘absolutely not’) to four points (‘a great deal’). The responses to the four items were then summed. A higher score indicated greater social support [34]. Three items were used to assess family relationship to support MMT use: ‘Are your family members familiar with MMT?’, ‘Do they support your participation in MMT?’ and ‘Did you communicate well with each other last month?’. Participants who responded with ‘agree or extremely agree’ were regarded as having good family awareness and support in their participation in MMT, as well as good MMT communication with family members.

#### 2.4.4. Health Status

The Short Form-12 (SF-12) was used to assess the health status of MMT clients. The Chinese version of the SF-12 index has demonstrated validity and reliability among various study populations in China [35,36]. This measure includes 12 self-rated items covering eight domains: general health (GH), physical functioning (PF), role physical (RP), bodily pain (BP), vitality (VT), social functioning (SF), role emotional (RE) and mental health (MH). The physical component summary (PCS) was determined by summing the scores of GH, PF, RP and BP. The mental component summary (MCS) was calculated by summing the remaining domain scores. A higher score indicated better health.

### 2.5. Sample Size Estimation

The primary outcome was the attrition rate at month six. According to previous studies, the estimated attrition rate in the control group at month six was 0.57 [37,38,39]. The smallest detectable between-group difference in the attrition rate was 10%. Assuming a 20% loss of sample due to adjustment by PSM [40], a sample of 173 individuals per arm would be needed to ensure a study power of at least 80%.

### 2.6. Statistical Analysis

As social workers were inclined to provide services to clients with certain characteristics, we conducted 1:1 matching according to the estimated probability of receiving the intervention (i.e., the propensity score, PS). We applied caliper matching, which only matches individuals with differences in PSs less than a caliper width equivalent to 0.2 times the standard deviation of the PS [41]. Subsequently, the standardized difference (SDiff) was used to assess the balances in covariates between arms [42]. The SDiff was calculated for each potential confounder and background variable before and after matching. A logistic regression model was then fitted to calculate the PS for each participant. According to a review on matching methods for studies with small sample sizes, the variables in the model used to calculate PS should be outcome-related and potentially able to yield covariate balance (SDiff < 0.25) after matching [43].

When fitting the PS model, based on the social workers’ knowledge and experience, we selected 10 baseline covariates from the participants’ baseline characteristics: marriage status, drug injection history, times of compulsory detoxifications, HIV infection, self-efficacy to retain in MMT in the next six months, willingness to accept social service, family support in clients’ participation in MMT, MMT communication with family members, PCS and MCS.

For categorical variables (including attrition), χ2 test was used to compare differences between groups, and relative risk (RR) was calculated. Cochran–Armitage test for trend was used to test whether the variables changed over time. Kaplan–Meier analysis was used to depict the survival curves of retention, which were compared between groups using the log-rank test. Mean differences in continuous variables were calculated. Analysis of variance for repeated measurement data was used to compare differences between groups at different timepoints. SPSS 20.0 [44], R 20.0 [45] and STATA 12 [46] were used for the data analysis.

## 3. Results

From July 2013 to April 2015, 202 clients were approached per arm. Of these, 186 and 173 consented to participate in the intervention and control groups, respectively. After PSM, 124 matched participants were included in each group.

### 3.1. Baseline Characteristics

In the original sample, participants in the intervention group were more likely than those in the control group to be female (14.5% versus 8.7%) and HIV positive (11.3% versus 1.7%), have ever injected drugs (73.1% versus 59.0%), have experienced ≥3 compulsory detoxifications (43.8% versus 33.9%) and be more willing to accept social services (87.6% versus 72.3%). Fewer clients in the intervention group reported having good MMT communication with family members (71.5% versus 80.9%). The intervention group also reported worse vitality (46.4±9.8 versus 49.0±9.2), worse role-emotional (37.8±21.0 versus 44.7±18.8) and worse mental health scores (40.4±13.0 versus 44.1±10.5) relative to the control group. After matching, no significant inter-group differences were observed in the baseline variables. In the matched sample, the majority of participants were male (88.3%), older than 45 years (58.5%), less educated (83.1% attended junior high or below), unemployed (59.7%), had ever injected drugs (64.9%), had experienced compulsory detoxification (78.1%), were currently on drugs (75.9%), did not have proper perceptions about the aim/duration of MMT (56.5%) or MMT dosage (57.7%), were eager to detoxify (37.5% reported a full intention score), were willing to accept psycho-social services (84.7%) and had a low or medium level of self-efficacy (72.6%). The baseline characteristics of the unmatched and matched samples are listed in Table 1.

### 3.2. Primary Outcome

The primary outcome was the attrition rate at month six (Table 2). The attrition rate was significantly lower in the intervention group than in the control group at month six (39.5% versus 52.4%, RR = 0.787, 95% CI: 0.623–0.993, *P* = 0.041). The absolute risk reduction at month six was 12.9%, indicating that interventions should be delivered to eight clients to avoid one case of attrition. At month one, the attrition rate was lower in the intervention group than in the control group (10.5% versus 19.4%, RR = 0.901, 95% CI: 0.811–1.001, *P* = 0.050). Similar results were reported at month 12 (58.9% versus 75.0%, RR = 0.608, 95% CI: 0.420–0.880, *P* = 0.007).

The retention curves are shown in Figure 2. A log-rank test revealed a higher retention rate in the intervention group throughout the study period. The difference in retention rates at month one was marginally significant (*P* = 0.056), while those at month 6 (*P* = 0.025) and month 12 (*P* = 0.005) were statistically significant.

### 3.3. Secondary Outcomes

Secondary outcomes are shown in Table 3. At month six, monthly income of the intervention group was significantly higher in the intervention group than in the control group (RR = 1.39, 95% CI: 1.04–1.85, *P* = 0.028). The detoxification intention score at month six was higher in the intervention group (RR = 1.40, 95% CI: 1.08–1.81, *P* = 0.012). The intervention group also reported a significantly higher level of family support in their participation in MMT (RR = 1.74, 95% CI: 1.07–2.82, *P* = 0.049). At month one, the inter-group difference in the employment status reached marginal significance (RR = 1.23, 95% CI: 0.99–1.51, *P* = 0.056). Regarding trends over time, the rates of self-reported current drug use decreased over time in both groups (control group: baseline, 73.4%; month one, 28.6%; month six, 21.6%; *P* < 0.001; intervention group: baseline, 78.2%; month one, 19.6%; month six, 13.2%; *P* < 0.001). In the intervention group, the detoxification intention score increased significantly over time [participants with full intention score (=10): baseline, 35.5%; month one, 49.1%; month six, 51.6%; *P* = 0.014]. An increasing trend in family support in clients’ participation in MMT was also observed (baseline, 64.5%; month one, 71.6%; month six, 77.8%; *P* < 0.001). However, the rates of proper perceptions about the MMT dosage also decreased significantly (participants with proper perceptions about MMT dosage: baseline, 46.0%; month one, 36.0%; month six, 29.5%; *P* = 0.012). No significant difference between inter was observed in other variables assessed at any time point or in other trends over time.

## 4. Discussion

Our study demonstrated that psycho-social services delivered by social workers to support MMT were effective in a real-world context, consistent with our previous findings in trial context. Based on the routine practices of social workers, we found that psycho-social services were provided to users with greater needs. To account for the imbalance between intervention and control groups, we used PSM and found that the intervention reduced attrition and improved the well-being of MMT clients in terms of monthly income, detoxification intention and family support. This study also highlighted the suboptimal retention in MMT, as demonstrated by the 6-month retention rates at 60.5% and 47.6% in the intervention and control groups, respectively. These findings corroborate MMT retention rates reported in other settings such as the US [47], Ireland [48] and Iran [49], characterizing an urgent need for improving MMT retention in those settings.

Beyond retention in MMT, this research demonstrated that the intervention improved several secondary outcomes including monthly income, detoxification intention and family support in clients’ participation in MMT. The monthly income at month six was higher in the intervention group, and this may be attributed to the higher employment rate at month six (employed participants: intervention group, 50%; control group, 37.7%). It may also be attributed to increased family support in the intervention group, given the crucial role of family members who provide financial support to drug-using clients in MMT clinics. Notably, social workers searched the local job market for suitable opportunities for intervention group participants and provided skill training to enhance the participants’ motivation and ability to restore their social roles. Additionally, the social workers provided customized family counseling sessions to help family members understand the importance of MMT to the clients. The 6-month data also corroborated this explanation, as 77.8% of intervention group participants reported family support in their participation in MMT, compared to only 61.4% of control group participants.

In both groups, the declines in self-reported current drug use over time demonstrate the effectiveness of MMT, with or without psycho-social services. Moreover, participants in the intervention group reported increasing detoxification intentions and family support, along with an unexpected decrease in the perception of MMT dosage. Apart from demonstrating the motivational and supportive effects of psycho-social services, this finding is noteworthy because it exposes an underlying contradiction between the clients’ wishes and the provided services. The combination of the clients’ increasing intention to detoxify and their worsening perceptions about MMT dosage allow us to infer safely that MMT clients strongly wish to be detoxified and free from any form of drugs including methadone. However, the psycho-social services provided in this study often emphasized the importance of sufficient MMT dosages, and thus might have dispelled the clients’ hopes of total abstinence and weakened the retention efficacy of the intervention. In the future, the delivery of correct information in a manner that sustains hope remains a challenging task for social workers. In retrospect, MMT has effectively reduced drug use in China since 2003 [50]. Daily MMT remains a mainstay in reducing heroin use and is recommended as an evidence-based intervention in many settings to people who use drugs. Our finding pointed to the need to increase sustained acceptability of MMT over time among people who inject drugs. Future study should explore the reasons and context of declined perceived needs of MMT, including MMT-related stigma, and thus, inform MMT-based interventions.

This study had several limitations. First, all participating clinics were located in the same city due to logistics reasons. This might undermine the generalizability of our findings to cities with distinct characteristics as compared to Guangzhou. However, we adopted a multi-center study design and included various study sites in Guangzhou. Second, the endpoint of previous studies varied and some have adopted a longer follow-up period such as one year to observe retention and attrition outcomes [51,52,53]. In the present study, we focused on 6-month outcomes, which may limit the comparability with previous studies with a different follow up time frame. However, results showed a steep decline of retention rate within the first six months which proved that six-month follow-up would be a reasonable observation period in this study context. Moreover, we measured and reported key findings at month one and month 12. Third, the intervention in this study was adjusted to adapt to the workload of social workers, which might reduce the effect of the psycho-social service. Despite such adaptation, the intervention still demonstrated the effectiveness in reducing attrition and improving psycho-social outcomes. Fourth, this study did not rule out the effect of underlying support from social workers. Future studies could test whether a friendly staff person’s regular chats and informal relationship with the clients can also increase retention. Lastly, the PSM method used in this study can be limited because only measured variables can be adjusted, which may lead to a risk of residual confounding after adjustment. Additionally, only subjects with PSs within the overlap of the two groups could be matched, which reduced the sample size and may have introduced bias in the matched sample [54]. Despite those limitations, PSM is widely used in circumstances where randomization is not possible or ethical, including real-world quasi-experimental studies such as the present study. By calculating and matching on PS, a synthesis of selected covariates, PSM simulates the effect of randomization to eliminate bias [55].

## 5. Conclusions

This study demonstrated the effectiveness of psycho-social services delivered by social workers when combined with standard MMT in a real-world context. The addition of the psycho-social services reduced attrition relative to MMT alone. This intervention proves to be feasible and effective, and our findings provided evidence-based support to psycho-social service provision as an integral part of practices at MMT clinics in China. Future research should gather evidence on potential benefits to expand psycho-social services to include all people who use drugs and generate greater impact.

## Figures and Tables

**Figure 1 ijerph-16-04337-f001:**
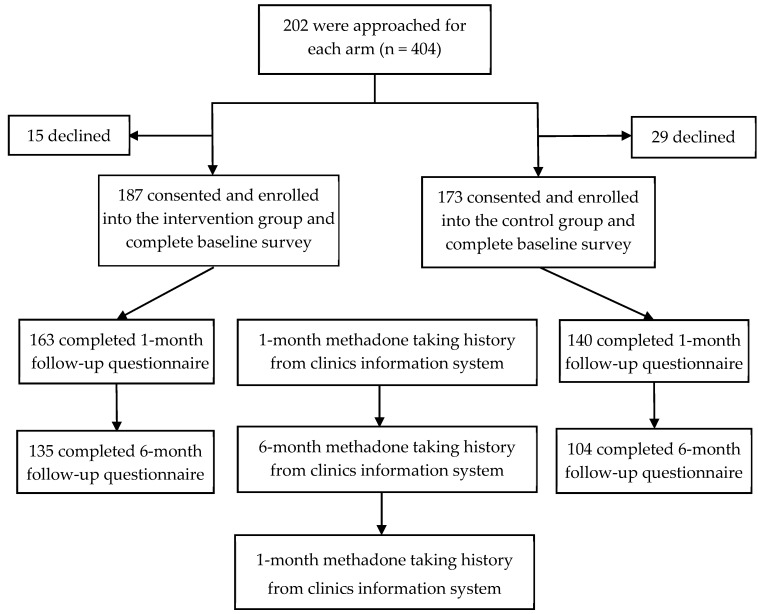
Flow chart of the study.

**Figure 2 ijerph-16-04337-f002:**
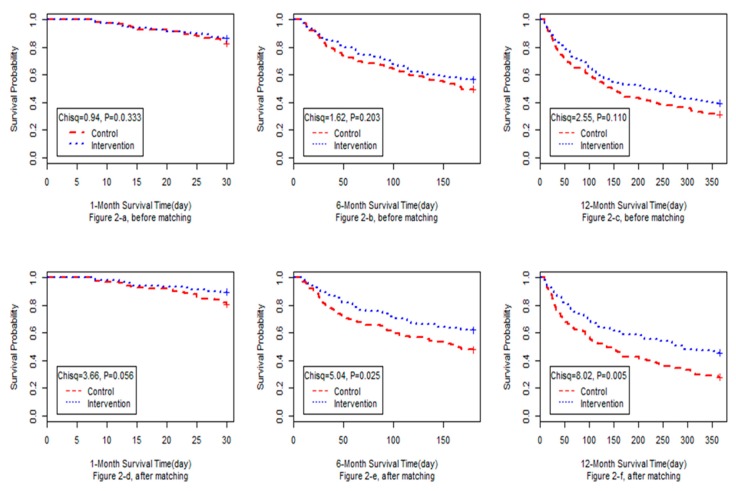
Retention curves of before-and after-matching sample: (**2-a**) 1-month retention curve of before-matching sample; (**2-b**) 6-month retention curve of before-matching sample; (**2-c**) 12-month retention curve of before-matching sample; (**2-d**) 1-month retention curve of after-matching sample; (**2-e**) 1-month retention curve of after-matching sample; (**2-f**) 1-month retention curve of after-matching sample.

**Table 1 ijerph-16-04337-t001:** Baseline characteristics of before-and after-matching sample.

	Before-matching Sample (n = 359)			After-matching Sample (n = 248)		
	Control	Intervention	SDiff	Control	Intervention	SDiff
	(%)	(%)		(%)	(%)	
**Socio-demographic characteristics**						
Sex (female)	8.7	14.5	0.182	10.5	12.9	0.075
Age (>45 years)	63.6	61.3	0.048	57.3	59.7	0.049
Education level						
Primary or below	17.3	21.0	0.094	17.7	19.4	0.044
Junior high	68.2	59.7	0.178	68.5	60.5	0.168
Senior high or above	14.5	19.4	0.131	13.7	20.2	0.174
Current marriage status						
Single	43.4	38.7	0.096	40.3	38.7	0.033
Married/cohabitating	40.5	39.8	0.014	44.4	41.9	0.050
Divorced/other	16.2	21.5	0.136	15.3	19.4	0.108
Current employment status (Employed)	33.5	40.3	0.141	35.5	45.2	0.199
Monthly income (>1000 CNY)	42.2	46.2	0.081	42.7	53.2	0.195
Being HIV positive (yes)	1.7	11.3	0.397^a^	2.4	4.0	0.091
**Drug use-related characteristics**						
Age of initial drug use (>25 years old)	60.1	54.3	0.117	56.5	54.0	0.050
Duration of drug use (>20 years)	58.4	61.6	0.065	56.5	59.3	0.057
Ever injected drugs (yes)	59.0	73.1	0.301^a^	65.3	64.5	0.017
Times of compulsory detoxification						
0	20.5	21.1	0.015	21.1	22.8	0.041
1-2	45.6	35.1	0.215	43.9	39.8	0.083
3~	33.9	43.8	0.204	35.0	37.4	0.050
Current drug use						
No use	20.2	21.0	0.020	26.6	21.8	0.112
Less than once a day	26.0	33.9	0.173	22.6	33.1	0.236
At least once a day	53.8	45.2	0.173	50.8	45.2	0.112
detox intention score						
<8	23.7	25.3	0.037	20.0	29.0	0.210
8-9	42.2	34.4	0.161	40.3	35.5	0.099
10	34.1	40.3	0.129	39.5	35.5	0.083
**MMT-related perceptions**						
Having proper perception about MMT aim/time	43.9	43.5	0.008	42.7	44.4	0.034
Having proper perception about MMT dosage	42.2	40.9	0.026	38.7	46.0	0.148
Self-efficacy of MMT retention in the next six months						
Low	24.3	26.9	0.060	25.0	25.0	<0.001
Medium	47.4	44.6	0.056	47.6	47.6	<0.001
High	28.3	28.5	0.004	27.4	27.4	<0.001
Willing to accept psycho-social services (yes)	72.3	87.6	0.389^a^	84.7	84.7	<0.001
**Family and Social Support**						
Score of Perceived Social Support Scale (Mean, SD)	10.4, 2.0	10.1, 2.4	0.149	10.2, 2.2	10.5, 2.2	0.147
Family awareness of MMT						
Disagree/Extremely disagree	21.4	29.6	0.189	21.8	21.0	0.020
Don’t know/hard to say	40.5	28.5	0.254^a^	36.3	33.1	0.067
Agree/Extremely agree	38.2	41.9	0.076	41.9	46.0	0.083
Family support in MMT						
Disagree/Extremely disagree	9.2	9.7	0.017	8.9	9.7	0.028
Don’t know/hard to say	28.3	27.4	0.020	25.8	25.8	<0.001
Agree/Extremely agree	62.4	62.9	0.010	65.3	64.5	0.017
Good MMT communication with family members						
Disagree/Extremely disagree	19.1	28.5	0.222	25.0	23.4	0.037
Don’t know/hard to say	40.5	26.3	0.305^a^	30.6	30.6	<0.001
Agree/Extremely agree	40.5	45.2	0.095	44.4	46.0	0.032
**Overall health status (Mean, SD)**						
Physical functioning	50.8, 9.0	49.4, 9.5	0.153	50.6, 9.1	50.7, 8.5	0.016
Role-physical	45.7, 16.5	44.9, 16.3	0.047	44.7, 17.0	46.9, 15.5	0.137
Role-emotional	44.7, 18.8	37.8, 21.0	0.346 ^a^	43.3, 19.6	43.5, 19.2	0.009
Mental health	45.9, 8.3	44.6, 10.3	0.147	46.3, 8.8	46.7, 9.2	0.049
General health	44.3, 9.4	43.2, 11.3	0.109	44.3, 9.9	42.8, 11.3	0.138
Bodily pain	49.8, 8.3	48.4, 9.0	0.160	50.1, 9.1	49.3, 8.3	0.095
Vitality	49.0, 9.2	46.4, 9.8	0.267 ^a^	49.0, 9.8	47.6, 9.3	0.150
Social functioning	43.3, 9.9	43.0, 10.5	0.033	42.6, 10.8	43.7, 9.7	0.114
Physical Component Summary	49.3, 7.5	49.2, 8.5	0.005	49.1, 7.8	49.2, 7.3	0.004
Mental Component Summary	44.1, 10.5	40.4, 13.0	0.320 ^a^	43.6, 11.0	43.7, 11.0	0.005

SDiff, standardized difference; CNY, Chinese Yuan; MMT, methadone maintenance treatment; SD, standard deviation; a, SDiff value > 0.25.

**Table 2 ijerph-16-04337-t002:** Attrition rates and relative risk of after-matching sample (N = 248).

Attrition Rates (%)	Propensity-Matched Cohort		
	M1	M6	M12
Control	19.4 (24/124)	52.4 (65/124)	75.0 (93/124)
Intervention	10.5 (13/124)	39.5 (49/124)	58.9 (73/124)
RR (95% CI)	0.901 (0.811,1.001)	0.787 (0.623,0.993) *	0.608 (0.420,0.880) *
Absolutely risk reduction (95% CI)	0.089 [0.001,0.177] *	0.129 (0.006,0.252) *	0.161 (0.046,0.277) *
Reciprocal of the absolute risk reduction	12	8	6

RR, relative risk; CI, confidence interval; *: *P* < 0.05.

**Table 3 ijerph-16-04337-t003:** Secondary outcomes of after-matching sample (N = 248).

	M0 (n = 248) n (%)	M1 (n = 211) n (%)	M6 (n = 172) n (%)	*P* for Trend
**Socio-economic status**				
Current employment status (Employed)				
Control	44 (35.5)	30 (30.3)	29 (37.7)	0.864
Intervention	56 (45.2)	47 (43.1)	47 (50.0)	0.516
RR (95% CI)	1.18 (0.96,1.45)	1.23 (0.99,1.51)^+^	1.25 (0.96,1.63)	
Monthly income (>1000 CNY)				
Control	54 (43.5)	38 (38.8)	30 (39.0)	0.481
Intervention	66 (53.2)	52 (47.7)	52 (55.9)	0.767
RR (95% CI)	1.21 (0.95,1.54)	1.17 (0.92,1.49)	1.39 (1.04,1.85)*	
**Drug use-related characteristics**				
Current drug use				
Control	91 (73.4)	26 (28.6)	16 (21.6)	<0.001*
Intervention	97 (78.2)	21 (19.6)	12 (13.2)	<0.001*
RR (95% CI)	1.22 (0.78,1.90)	0.889 (0.76,1.04)	0.903 (0.78,1.04)	
Detox intention score (=10)				
Control	49 (39.5)	42 (42.4)	25 (32.5)	0.391
Intervention	44 (35.5)	54 (49.1)	49 (51.6)	0.014*
RR (95% CI)	0.94 (0.77,1.14)	1.13 (0.88,1.45)	1.40 (1.08,1.81)*	
**MMT-related perceptions**				
Proper perception about MMT aim/duration				
Control	53 (42.7)	50 (50.0)	34 (44.2)	0.730
Intervention	55 (44.4)	53 (47.7)	52 (54.7)	0.133
RR (95% CI)	1.03 (0.83,1.28)	0.96 (0.73,1.25)	1.23 (0.92,1.66)	
Proper perception about MMT dosage				
Control	48 (38.7)	39 (39.0)	28 (36.4)	0.763
Intervention	57 (46.0)	40 (36.0)	28 (29.5)	0.012*
RR (95% CI)	1.13 (0.92,1.41)	0.95 (0.77,1.18)	0.90 (0.73,1.12)	
**Social Support**				
Score of Perceived Social Support Scale (Mean, SD)				
Control	10.20,2.19	10.39,2.57	10.30,1.99	0.887
Intervention	10.52,2.18	10.22,2.35	10.08,2.16	0.059^+^
Mean diff (95% CI)	0.28 (−0.22,0.87)	−0.03 (−0.69,0.63)	−0.23 (−0.86,0.41)	
Family awareness of MMT (agree/extremely agree)				
Control	52 (41.9)	55 (55.6)	31 (44.3)	0.513
Intervention	57 (46.0)	63 (57.3)	49 (54.4)	0.180
RR (95% CI)	1.08 (0.86,1.34)	1.04 (0.76,1.42)	1.22 (0.90,1.66)	
Family support in MMT (Agree/Extremely agree)				
Control	81 (65.3)	71 (71.7)	43 (61.4)	0.748
Intervention	80 (64.5)	78 (71.6)	70 (77.8)	0.034*
RR (95% CI)	0.98 (0.70,1.37)	0.99 (0.65,1.53)	1.74 (1.07,2.82) *	
Good MMT communication with family members (Agree/Extremely agree)				
Control	55 (44.4)	54 (54.5)	28 (40.0)	0.796
Intervention	57 (46.0)	60 (54.5)	46 (51.1)	0.401
RR (95% CI)	1.03 (0.82,1.29)	1.00 (0.74,1.35)	1.23 (0.92,1.63)	
**Overall health status (Mean, SD)**				
Physical component summary				
Control	49.13,7.81	50.60,7.61	50.36,7.93	0.717
Intervention	49.16,7.33	51.40,6.99	50.17,8.05	0.165
Mean difference (95% CI)	0.03 (−1.86,1.93)	0.80 (−1.18,2.79)	−0.20 (−2.62,2.23)	
Mental component summary				
Control	43.64,11.01	45.07,11.13	43.70,11.10	0.396
Intervention	43.70,10.98	42.93,11.68	46.46,12.47	0.058^+^
Mean difference (95% CI)	0.06 (−2.69,2.81)	−2.14 (−5.25,0.97)	2.77 (−0.80,6.33)	

RR, relative risk; CI, confidence interval; CNY, Chinese Yuan; MMT, methadone maintenance treatment; SD, standard deviation; ^+^, *P* < 0.1; *, *P* < 0.05.

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
