# Peer review of "Effectiveness of a Psycho-Social Intervention Aimed at Reducing Attrition at Methadone Maintenance Treatment Clinics: A Propensity Score Matching Analysis"

_ijerph, 2019, doi:10.3390/ijerph16224337_

Round 1
Reviewer 1 Report
Overall a nice study. Addresses significant challenges in successful treatment of population described. An important social issue in many parts of the world. Methodological challenges in conducting this type of research dealt with in convincing manner. Limitations addressed. Adds to the literature regarding fine tuning treatment regimens to increase efficiency and enhance outcomes. A few specifics below
line 45-is there more current data to add/
line 54 'same as above' to add to ref. 14.
line 86- sentence awkward please reword.
line 105-sentence awkward please reword
Author Response
Dear reviewer, thank you for your recognition. We are very delighted to receive your comment and suggestion. The reference data was updated and wordings were rephrased discreetly upon receiving your review. Following are the details of the revision made in accordance with your suggestion.
Point 1: line 45-is there more current data to add.
Response 1: We have updated with current data as suggested. (Line 44-45 of the revised version); Reference [1] was replaced. (Line 378-379 of the revised version).
Point 2: line 54 'same as above' to add to ref. 14.
Response 2: We have updated Reference 14 with a recent one, which was a six-year longitudinal study in China to reflect the retention problem in Chinese MMT clinics. Reference [14] was replaced. (Line 414-416 of the revised version).
Point 3: line 86- sentence awkward please reword.
Response 3: The sentence was rephrased into “This study aimed to evaluate the effectiveness of the services in real-world settings, following the demonstration of the efficacy in the previous RCT. Quasi-Experimental study design and propensity score matching (PSM) were used to evaluated differences in attrition and psycho-social status after balancing potential confounding factors between the intervention and control groups.” (Line 87-90 of the revised version).
Point 4: line 105-sentence awkward please reword.
Response 4: The sentence was rephrased into “…after recruiting one participant to the intervention group, the social worker invited the next client attending the clinic, who does not have the prioritized characteristics, to the control group. The 6-month psycho-social services were delivered to MMT clients of the intervention group by social workers after their enrolment. (Line 109-112 of the revised version).
Reviewer 2 Report
Thank you for the opportunity to review ‘Effectiveness of a psycho-social intervention aimed at reducing attrition at methadone maintenance treatment clinics: A propensity score matching analysis’. This was a well-designed study and beautifully written paper. I have no suggestions on the way the paper could be better written: it was a pleasure to read in every way. The Introduction sets out the context of the study, the Materials and Methods section was set out in excellent clarity, the Results section well summarised, statistics appropriate, and the Discussion and Conclusions sections well-founded on the data presented. I usually have a lot to say about papers I review, but this time there is no need as I think the authors have done a superlative piece of work.
That said, there are two things I would encourage the authors to consider. Firstly, the word ‘proved’ or ‘proving’ (Abstract line 36 and p. 12 line 312) I think is significantly overclaiming. As responsible scientists, as the authors clearly are, they will know that it is just about impossible to ‘prove’ anything. I strongly recommend the words ‘demonstrated’ and ‘demonstrating’ in these two cases.
On a more substantive note, I think that the authors need to look at the notion of psychosocial intervention. As a social worker I would love think that my profession had such a positive impact. However, there is a very real possibility in this study that the simple action of engaging and relationship-building with clients (rather than some professional intervention) was sufficient to keep them engaged with the MMT service over a long period of time. This could be noted as a limitation to the study, as the authors did not test for this. The authors may wish to confirm their findings by designing a similar study where a third intervention option is just a friendly staff person who regularly chats and builds an informal relationship with the clients. This would test whether retention was based on the psychosocial intervention or the mere fact of having a relationship with someone in the clinic.
Otherwise I congratulate the authors on a very fine paper.
Author Response
Dear reviewer, thank you for your highly speaking of this manuscript. We are very delighted to receive your comments and suggestions. Following are the details of the revision made in accordance with your suggestion.
Point 1: Firstly, the word ‘proved’ or ‘proving’ (Abstract line 36 and p. 12 line 312) I think is significantly overclaiming. As responsible scientists, as the authors clearly are, they will know that it is just about impossible to ‘prove’ anything. I strongly recommend the words ‘demonstrated’ and ‘demonstrating’ in these two cases.
Response 1: The wordings were changed accordingly as suggested. (Line 37 & 321 of the revised version)
Point 2: On a more substantive note, I think that the authors need to look at the notion of psychosocial intervention. As a social worker, I would love to think that my profession had such a positive impact. However, there is a very real possibility in this study that the simple action of engaging and relationship-building with clients (rather than some professional intervention) was sufficient to keep them engaged with the MMT service over a long period of time. This could be noted as a limitation to the study, as the authors did not test for this. The authors may wish to confirm their findings by designing a similar study where a third intervention option is just a friendly staff person who regularly chats and builds an informal relationship with the clients. This would test whether retention was based on the psychosocial intervention or the mere fact of having a relationship with someone in the clinic.
Response 2: Thank you for the comments. We agree that in our study the effectiveness of engaging and relationship-building with clients was combined with that of the professional intervention. We have added this as a limitation (Line 348-351 of the revised version). We also agree that we should test whether a friendly staff person’s regular chats and informal relationship with the clients can also increase retention. If it is effective, that will be a cost-effective simple service in MMT, especially in resource-limited areas.
The following sentence was added to the limitation part:
“Fourth, this study did not rule out the effect of underlying support from social workers. Future studies could test whether a friendly staff person’s regular chats and informal relationship with the clients can also increase retention.” (Line 348-351 of the revised version)